mathematical modelling/behaviour/evolution

evolutionary games, expected utility theory, social dilemma, social viscosity, altruism

**Author for correspondence:**
Hiromu Ito
e-mail: ito.hiromu@nagasaki-u.ac.jp

# Dynamic utility: the sixth reciprocity mechanism for the evolution of cooperation

Hiromu Ito[1] and Jun Tanimoto[2,3]

[1]Department of International Health and Medical Anthropology, Institute of Tropical Medicine, Nagasaki University, Nagasaki 852-8523, Japan
[2]Department of Energy and Environmental Engineering, Interdisciplinary Graduate School of Engineering Sciences, and [3]Department of Advanced Environmental Science and Engineering, Faculty of Engineering Sciences, Kyushu University, Fukuoka 816-8580, Japan

HI, 0000-0001-9350-0546

Game theory has been extensively applied to elucidate the evolutionary mechanism of cooperative behaviour. Dilemmas in game theory are important elements that disturb the promotion of cooperation. An important question is how to escape from dilemmas. Recently, a dynamic utility function (DUF) that considers an individual's current status (wealth) and that can be applied to game theory was developed. The DUF is different from the famous five reciprocity mechanisms called Nowak's five rules. Under the DUF, cooperation is promoted by poor players in the chicken game, with no changes in the prisoner's dilemma and stag-hunt games. In this paper, by comparing the strengths of the two dilemmas, we show that the DUF is a novel reciprocity mechanism (sixth rule) that differs from Nowak's five rules. We also show the difference in dilemma relaxation between dynamic game theory and (traditional) static game theory when the DUF and one of the five rules are combined. Our results indicate that poor players unequivocally promote cooperation in any dynamic game. Unlike conventional rules that have to be brought into game settings, this sixth rule is universally (canonical form) applicable to any game because all repeated/evolutionary games are dynamic in principle.

## 1. Background

The evolution of cooperation in human and animal societies is enigmatic because a non-cooperative agent (defector) can obtain an evolutionarily selective advantage by taking the benefits of social contributions of other cooperators while avoiding the costs of cooperation [1]. However, we often observe cooperative behaviour in human and animal societies, even though society is constructed by non-kin agents [2,3]. Game theory has been extensively studied to explain how cooperation is promoted in human and animal

societies [4–9]. One of the main foci of studies in game theory is the kind of reciprocity mechanisms that can resolve social dilemmas that disturb the promotion and evolution of cooperative behaviour and how the reciprocity mechanisms can allow players to escape from dilemmas [9–11]. In game theory, many $2 \times 2$ (pairwise) dilemma games have been built to investigate the types of reciprocity mechanisms that enable a player to overcome conflicts of interests and promote cooperative behaviour [10,11]. We can denote the pay-off matrix of pairwise games with two strategies: cooperation (C) and defection (D). The rewards of players are determined by the pay-off matrix and the strategies that the players choose (equation (1.1)).

$$A \equiv [a_{ij}] = \begin{array}{c} \\ C \\ D \end{array} \begin{array}{cc} C & D \\ \begin{pmatrix} R & S \\ T & P \end{pmatrix} \end{array}. \tag{1.1}$$

This pay-off matrix means as follows: if both cooperate, they receive the 'reward' $R$; if both defect, they get 'punishment' $P$; and if one chooses cooperation while the other defects, the defector gets the 'temptation' $T$ and the cooperator left the pay-off of 'sucker' $S$ [11].

In a pairwise game, there are two indicators with which to measure the strength of the dilemma situation. One is the gamble-intending dilemma (GID), which appears because players try to exploit their opponents, and the other is the risk-averting dilemma (RAD), which appears because players try not to be exploited by their opponents [9,12–14]. The strengths of these two dilemmas, namely, the GID and RAD, can be calculated from the elements of the pay-off matrix (equation (1.1)) [14]. Let $D'_g$ and $D_r'$ be the values of GID and RAD, respectively. Then, we obtain the following:

$$D'_g = \frac{T - R}{R - P} \tag{1.2}$$

and

$$D'_r = \frac{P - S}{R - P}. \tag{1.3}$$

Note that the following equations are established by defining [15]

$$T = R + (R - P)D'_g \tag{1.4}$$

and

$$S = P - (R - P)D'_r. \tag{1.5}$$

Depending on the strengths of these two dilemmas, the game can be divided into four classes: a prisoner's dilemma (PD) game, a chicken game (also known as a snowdrift or hawk-dove game), a stag-hunt (SH) game and a trivial game with no dilemma. Therefore, we can evaluate the evolution of cooperation more precisely if we quantitatively compare the two constitutional strengths of the reciprocity mechanisms in all pairwise games (irrespective of the reciprocity mechanisms and finiteness properties) using a RAD–GID phase plane diagram that consists of the two standardized measures (figure 1*a*) [15]. According to the concept of universal scaling, the relaxation of these two types of dilemmas is expressed by shifting the x-axis (i.e. the RAD-axis) and the y-axis (i.e. the GID-axis) of the RAD–GID phase plane diagram to the positive domain [15]. In this paper, we refer to the $D'_r - D'_g$ phase diagram without reciprocity as the 'default' (figure 1*a*). Note that in the RAD–GID phase diagram, the first, second, third and fourth quadrants represent the PD, chicken, trivial and SH game structures, respectively (figure 1*a*).

Nowak's five reciprocity rules (i.e. direct reciprocity, indirect reciprocity, kin selection, group selection and network reciprocity) work as reciprocity mechanisms to resolve (relax) social dilemmas and promote cooperative behaviour [9,14,16]. These fundamental mechanisms are collectively known as social viscosity. In our previous paper, we used a RAD–GID phase plane diagram to visually show how Nowak's five rules relax the dilemma structure [15]. Our results showed that Nowak's five rules had different relaxation functions for the two dilemma strengths [15].

Recently, however, the promotion of cooperation was reported in a chicken game using a dynamic utility function (DUF) [17]. The DUF was developed in dynamic utility theory based on the maximization of a stochastic growth process by applying the optimality principle of Bellman's dynamic programming [18,19]. Because dynamic utility theory optimizes Markov chains (stochastic processes) as a form of sequential decision making, it maximizes the geometric mean of multiplicative growth rates [20]. The DUF is derived as follows [21,22]. Let time $t = 0, \ldots, T$ (final time), and let $w_t$ and $r_t$ represent wealth and the growth rate, respectively, at time $t$. Note that $r_t$ (>0) is the non-negative state variable of a decision maker

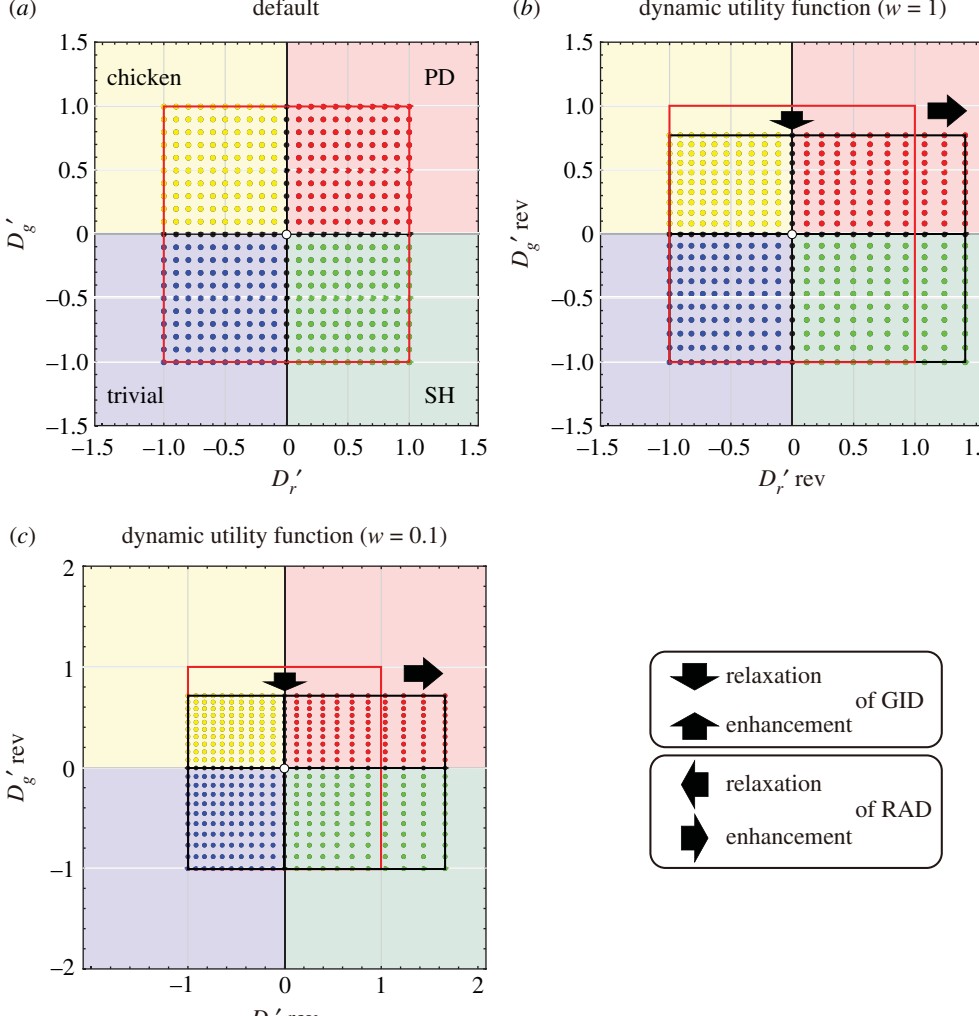

**Figure 1.** Phase planes of a pairwise game with coordinate movements with the introduction of a dynamic utility function. The shaded background colours indicate the regions of trivial (blue), prisoner's dilemma (PD) (red), chicken (yellow) and stag-hunt (SH) (green) games. The default game classes for each coordinate are indicated by the colour of the dots. (a) Default RAD–GID phase plane. (b,c) Transformed phase plane after the introduction of the DUF with $R = 3$ and $P = 2$. The current wealth of the player is (b) $w = 1$ and (c) $w = 0.1$. The red squares indicate the default phase planes. The thick black arrows indicate the relaxation and enhancement of the GID and RAD.

(independent, identically distributed random variable). Let $r_t$ denote the multiplicative growth rate of wealth at time $t$, such that $w_{t+1} = r_t w_t$. Wealth at the final time point, $w_T$, is then expressed as follows: $w_T = w_0 r_0 r_1 r_2 \cdots r_{T-1}$. We assume that the growth rates $r_t$ ($t = 0, \ldots, T$) are independent, identically distributed random variables that represent a stochastic process. The decision maker can optimize this stochastic process by choosing the best option at every time point. Therefore, we maximize wealth at the final time point $T$, $w_T$, such that $w_T \to \max$. The maximization of $w_T$ is equivalent to that of the geometric mean growth rate, such that $G(r) = \prod_{i=0}^{T-1} r_i^{1/T} \to \max$. Taking the logarithm, we obtain $\log\{G(r)\} = 1/T \sum \log(r_i) = \mathrm{E}\{\log(r)\} : \to \max$. Therefore, we can define the DUF $u(r)$ for this maximization as $u(r) = \log r$. We now maximize the expected dynamic utility $\mathrm{E}(u)$ [23]. From the temporal equation $w_{t+1} = r_t w_t$, we can rewrite $r_t = w_{t+1}/w_t = (g_t + w_t)/w_t$ to obtain $r = (g + w)/w$, where $g$ and $w$ are the current gain and current wealth, respectively. The growth utility formula is then rewritten in the form of $g$ (decision variable) given $w$ (state variable), such that

$$u(g; w) = \log\left(\frac{g + w}{w}\right), \tag{1.6}$$

and we maximize the expected utility $\mathrm{E}\{u(g; w)\}$, which indicates that current wealth is the state variable for maximization of final wealth. Thus, the derived dynamic utility is in the form of a logarithmic function

(equation (1.6)). Note that the value of $g$ satisfies $-w < g$. This analytical solution demonstrates that the utility function depends on the current gain (decision variable) and the current wealth status (state variable) at the time of decision making, as in dynamic programming [17–24]. These properties show that cooperative behaviour evolves with the introduction of a DUF that accommodates the current wealth condition of individuals (decision makers) without the known five reciprocity mechanisms. However, we cannot explain why cooperation is promoted by poor players (whose current wealth, $w$, are very low) in the chicken game but not in the PD game or SH game [17].

Here, we combine two new developments, namely, universal scaling parameters and the DUF. Specifically, we apply the DUF to a traditional (well-mixed infinite population) $2 \times 2$ game and analyse how it relaxes the strengths of the two dilemmas. By drawing the RAD–GID phase plane diagram, we compare the dilemma relaxation mechanism of the DUF with that of the Nowak's five rules which were investigated in a previous study [15]. According to this comparison, we present that the DUF has an entirely different dilemma relaxation mechanism from that of the all five rules. Then, we show the dilemma relaxation when the DUF and one of the five rules are combined. We here introduce the completely different pictures from previous studies that have achieved only an understanding of the social dilemma relaxation mechanism of the five rules [15]. Our aim is to demonstrate the difference between dynamic game theory and (traditional) static game theory. Finally, we discuss and predict the evolution of cooperation in truly dynamic games.

## 2. Methods

Here, we verify the two dilemma strengths of a $2 \times 2$ dynamic game comprising a pay-off matrix (equation (1.1)). We assume an infinite, well-mixed population (i.e. an infinite number of agents) with no previous social viscosities. Two individuals (players) are selected from an unlimited population at random and asked to play the game. Players receive a reward depending on the selected strategies C and D (equation (1.1)).

### 2.1. Dynamic utility function

Here, we introduce the concept of individual current status from the DUF, where $w$ is the current wealth of the player:

$$\begin{pmatrix} \log\left(\dfrac{R + w}{w}\right) & \log\left(\dfrac{S + w}{w}\right) \\ \log\left(\dfrac{T + w}{w}\right) & \log\left(\dfrac{P + w}{w}\right) \end{pmatrix}. \tag{2.1}$$

Then, the coordinates $(D'_r, D'_g)$ are transformed to $(D'_{r\,\text{rev}}, D'_{g\,\text{rev}})$ by the DUF as follows:

$$D'_{g\,\text{rev}} = \frac{\log\left((T+w)/w\right) - \log\left((R+w)/w\right)}{\log\left((R+w)/w\right) - \log\left((P+w)/w\right)} = \frac{\log\left[(T+w)/(R+w)\right]}{\log\left[(R+w)/(P+w)\right]} = f_{\text{DUF}}(T,R,P) = f_{\text{DUF}}(D'_g,R,P) \tag{2.2}$$

and

$$D'_{r\,\text{rev}} = \frac{\log\left((P+w)/w\right) - \log\left((S+w)/w\right)}{\log\left((R+w)/w\right) - \log\left((P+w)/w\right)} = \frac{\log\left[(P+w)/(S+w)\right]}{\log\left[(R+w)/(P+w)\right]} = g_{\text{DUF}}(S,R,P) = g_{\text{DUF}}(D'_r,R,P), \tag{2.3}$$

when $R = 3$ and $P = 2$, $T$ and $S$ are derived from equations (1.4) and (1.5) based on the values of $D'_r$ and $D'_g$. Therefore, when $w = 1$, the coordinates $(D'_r, D'_g) = \{(1, 1), (-1, 1), (-1, -1)$ and $(1, -1)\}$ shift to $(D'_{r\,\text{rev}}, D'_{g\,\text{rev}}) = \{(1.41, 0.78), (-1, 0.78), (-1, -1)$ and $(1.41, -1)\}$ by means of the DUF, respectively (figure 1b).

By introducing dynamic utility, i.e. the sixth reciprocity mechanism, the coordinates $(D'_r, D'_g)$ of the default game are transferred to the new coordinates $(D'_{r\,\text{rev}}, D'_{g\,\text{rev}})$ (figure 1). The DUF-transformed diagram depends on the current wealth level $w$ (figure 1b and c: $w$-dependence of the new diagram). Mathematically, if current wealth $w$ is small, the distortion of the dilemma structure is very large compared with the static model. But when $w \to \infty$ (means a person becomes extraordinarily rich), this

distortion disappears and converges to the solution for the static model

$$D'_{g\,rev} = \lim_{w\to\infty}\left[\frac{\log\left[(T+w)/(R+w)\right]}{\log\left[(R+w)/(P+w)\right]}\right] = \lim_{w\to\infty}\left[\frac{\log\left(1+T/w\right)-\log\left(1+R/w\right)}{\log\left(1+R/w\right)-\log\left(1+P/w\right)}\right] \cong \frac{T/w-R/w}{R/w-P/w}$$

$$= \frac{T-R}{R-P} = D'_g \qquad (2.4)$$

and

$$D'_{r\,rev} = \lim_{w\to\infty}\left[\frac{\log\left[(P+w)/(S+w)\right]}{\log\left[(R+w)/(P+w)\right]}\right] = \lim_{w\to\infty}\left[\frac{\log\left(1+P/w\right)-\log\left(1+S/w\right)}{\log\left(1+R/w\right)-\log\left(1+P/w\right)}\right] \cong \frac{P/w-S/w}{R/w-P/w} = \frac{P-S}{R-P}$$

$$= D'_r. \qquad (2.5)$$

Thus, the static model is considered an approximate model of the DUF games for extremely rich people, but not for the ordinary people.

Note that the default phase plane is the region enclosed by the red square. The default game classes for each coordinate are indicated by the colour of the dot. For example, the red dot is the coordinate of the default PD game without any reciprocity mechanism. The introduction of reciprocity mechanisms enhances or relaxes the strengths of the two dilemmas, and the phase plane is transformed into the black square from the red default square. The transformation of the coordinates changes the game class in the region where the dot moved from the same coloured background to another coloured background. For example, if a red dot moves to a region with a green background, the game structure in that region has changed from a PD game to an SH game due to the introduction of the reciprocity mechanism. These methods expand upon those detailed within our previous work [15]. Note that the introduction of the DUF does not change the game class.

## 2.2. Direct reciprocity

$$\begin{pmatrix} \dfrac{R}{1-\lambda} & S+\dfrac{\lambda P}{1-\lambda} \\ T+\dfrac{\lambda P}{1-\lambda} & \dfrac{P}{1-\lambda} \end{pmatrix}. \qquad (2.6)$$

Note that $\lambda$ is the probability of two players meeting each other in another round. The coordinates $(D'_r, D'_g)$ are transferred to $(D'_{r\,rev}, D'_{g\,rev})$ by direct reciprocity as follows:

$$D'_{g\,rev} = \frac{(T+(\lambda P/1-\lambda))-(R/(1-\lambda))}{(R/(1-\lambda))-(P/(1-\lambda))} = f_{DR}(T,R,P) = f_{DR}(D'_g,R,P) \qquad (2.7)$$

and

$$D'_{r\,rev} = \frac{\left(\dfrac{P}{1-\lambda}\right)-\left(S+\dfrac{\lambda P}{1-\lambda}\right)}{\left(\dfrac{R}{1-\lambda}\right)-\left(\dfrac{P}{1-\lambda}\right)} = g_{DR}(S,R,P) = g_{DR}(D'_r,R,P) \qquad (2.8)$$

when $R = 3$ and $P = 2$, $T$ and $S$ are derived from equations (1.4) and (1.5) based on the values of $D'_r$ and $D'_g$. Therefore, when $\lambda = 0.2$, the coordinate $(D'_r, D'_g) = (1, 1)$ shifts to $(D'_{r\,rev}, D'_{g\,rev}) = (0.8, 0.6)$ by means of direct reciprocity (figure 2a). Here, we also show the derivations of the dilemma strength of a game that applies one of the other four reciprocity rules (i.e. indirect reciprocity, kin selection, group selection and network reciprocity).

## 2.3. Indirect reciprocity

$$\begin{pmatrix} R & (1-q)S+qP \\ (1-q)T+qP & P \end{pmatrix}. \qquad (2.9)$$

Note that $q$ is the probability of knowing the reputation of another individual. The coordinates $(D'_r, D'_g)$ are transferred to $(D'_{r\,rev}, D'_{g\,rev})$ by indirect reciprocity as follows (figure 2a):

$$D'_{g\,rev} = \frac{\{(1-q)T+qP\}-R}{R-P} = f_{IR}(T,R,P) = f_{IR}(D'_g,R,P) \qquad (2.10)$$

and

$$D'_{r\,rev} = \frac{\{(1-q)T+qP\}-\{(1-q)S+qP\}}{R-P} = g_{IR}(S,R,P) = g_{IR}(D'_r,R,P). \qquad (2.11)$$

Note that direct reciprocity and indirect reciprocity of the same strength will transfer the phase plane to the same coordinates [25].

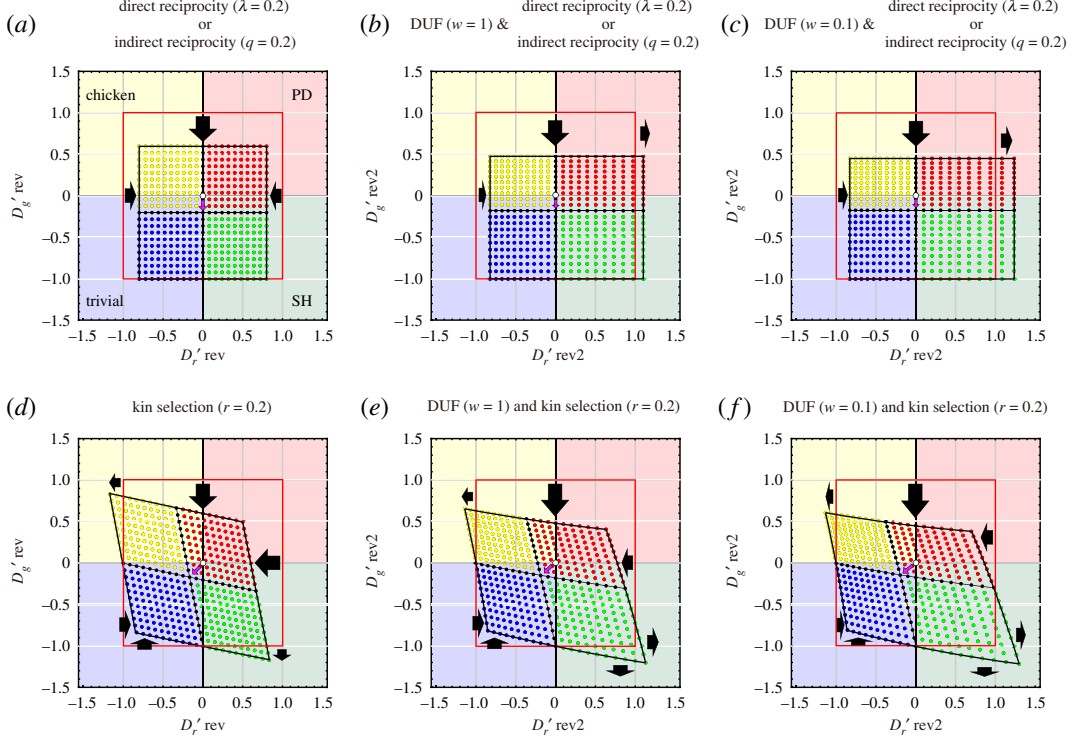

**Figure 2.** Transformation of the default phase plane ($-1 \leq D'_g, D'_r \leq +1$) with the introduction of the DUF and three rules (direct reciprocity, indirect reciprocity and kin selection) in all $2 \times 2$ games. The shaded background colours indicate the regions of trivial (blue), prisoner's dilemma (PD) (red), chicken (yellow) and stag-hunt (SH) (green) games. The default game classes for each coordinate are indicated by the colour of the dots. Transformed phase plane with the introduction of (*a–c*) direct reciprocity or indirect reciprocity and (*d–f*) kin selection. Direct reciprocity and indirect reciprocity both transfer the phase plane to the same coordinates. (*b,e*) The DUF for the rich player ($w = 1$) and (*c,f*) that for the poor player ($w = 0.1$) are introduced. The origin in the default (white circle) moves in the direction of the points indicated by pink arrows. The red squares indicate the default phase planes. The thick black arrows indicate the relaxation and enhancement of the GID and RAD. The basic parameters of the pay-off matrix are fixed as $R = 3$ and $P = 2$.

## 2.4. Kin selection

$$\begin{pmatrix} R & \dfrac{S+rT}{1+r} \\ \dfrac{T+rS}{1+r} & P \end{pmatrix}. \tag{2.12}$$

Note that $r$ is the average relatedness between interacting individuals. The coordinates $(D'_r, D'_g)$ are transferred to $(D'_{r\,\text{rev}}, D'_{g\,\text{rev}})$ by kin selection as follows (figure 2*d*):

$$D'_{g\,\text{rev}} = \frac{\left(\dfrac{T+rS}{1+r}\right) - R}{R - P} = f_{KS}(T,R,P) = f_{KS}(D'_g,R,P) \tag{2.13}$$

and

$$D'_{r\,\text{rev}} = \frac{P - \left(\dfrac{S+rT}{1+r}\right)}{R - P} = g_{KS}(S,R,P) = g_{KS}(D'_r,R,P). \tag{2.14}$$

## 2.5. Group selection

$$\begin{pmatrix} (m+n)R & nS+mR \\ nT+mP & (m+n)P \end{pmatrix}. \tag{2.15}$$

Note that $m$ is the number of groups and $n$ is the maximum size of a group. The coordinates $(D'_r, D'_g)$ are transferred to $(D'_{r\,\text{rev}}, D'_{g\,\text{rev}})$ by group selection as follows (figure 3*a*):

$$D'_{g\,\text{rev}} = \frac{(nT+mP) - (m+n)R}{(m+n)R - (m+n)P} = f_{GS}(T,R,P) = f_{GS}(D'_g,R,P) \tag{2.16}$$

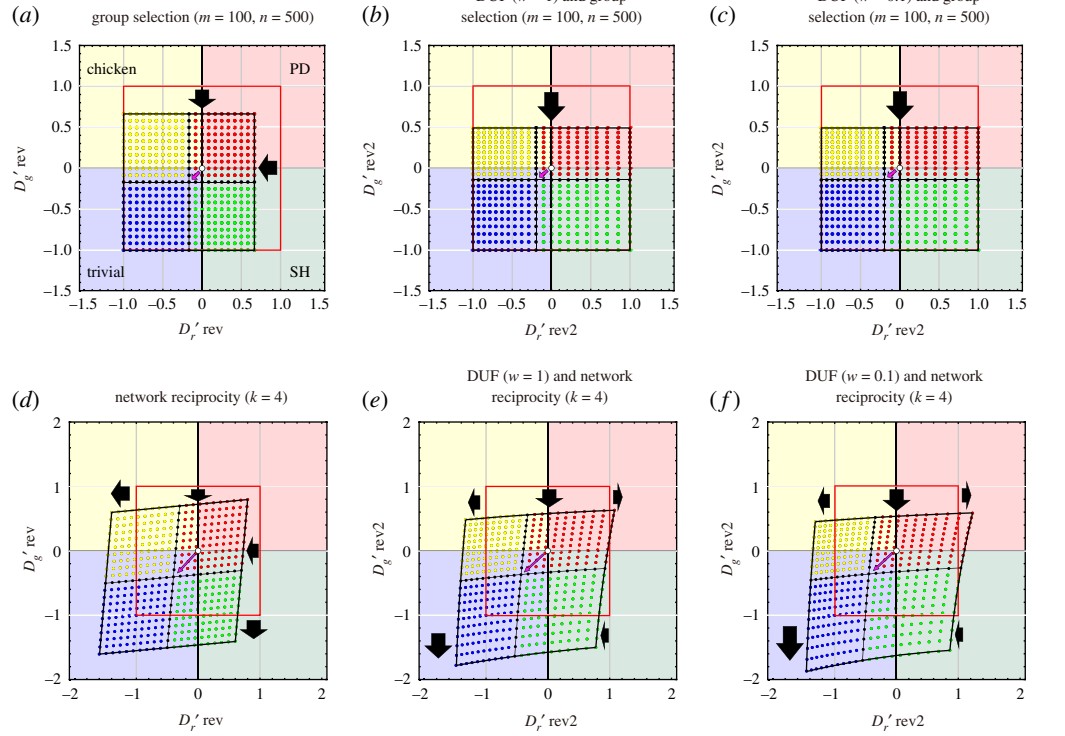

**Figure 3.** Transformation of the default phase plane $(-1 \leq D'_g, D'_r \leq +1)$ with the introduction of the DUF and group selection or network reciprocity in all $2 \times 2$ games. The shaded background colours indicate the regions of trivial (blue), prisoner's dilemma (PD) (red), chicken (yellow) and stag-hunt (SH) (green) games. The default game classes for each coordinate are indicated by the colour of the dots. Transformed phase plane with the introduction of $(a–c)$ group selection and $(d–f)$ network reciprocity. $(b,e)$ The DUF for the rich player $(w = 1)$ and $(c,f)$ that for the poor player $(w = 0.1)$ are introduced. The origin in the default (white circle) moves in the direction of the points indicated by pink arrows. The red squares indicate the default phase planes. The thick black arrows indicate the relaxation and enhancement of the GID and RAD. The basic parameters of the pay-off matrix are fixed as $R = 3$ and $P = 2$.

and

$$D'_{r\,\mathrm{rev}} = \frac{(m+n)P - (nS + mR)}{(m+n)R - (m+n)P} = g_{GS}(S,R,P) = g_{GS}(D'_r,R,P). \tag{2.17}$$

## 2.6. Network reciprocity

$$\begin{pmatrix} R & S+H \\ T-H & P \end{pmatrix}. \tag{2.18}$$

The term $H$ is defined as follows:

$$H = \frac{(k+1)(R-P) - T + S}{(k+1)(k-2)}. \tag{2.19}$$

Note that $k$ is the number of neighbours. The coordinates $(D'_r, D'_g)$ are transferred to $(D'_{r\,\mathrm{rev}}, D'_{g\,\mathrm{rev}})$ by direct reciprocity as follows (figure 3$d$):

$$D'_{g\,\mathrm{rev}} = \frac{(T-H) - R}{R-P} = f_{NR}(T,R,P) = f_{NR}(D'_g,R,P) \tag{2.20}$$

and

$$D'_{r\,\mathrm{rev}} = \frac{P - (S+H)}{R-P} = g_{NR}(S,R,P) = g_{NR}(D'_r,R,P). \tag{2.21}$$

$D'_{r\,\mathrm{rev}}$ and $D'_{g\,\mathrm{rev}}$ are derived according to equations (2.2)–(2.3) (DUF), equations (2.7)–(2.8) (direct reciprocity), equations (2.10)–(2.11) (indirect reciprocity), equations (2.13)–(2.14) (kin selection), equations (2.16)–(2.17) (group selection) or equations (2.20)–(2.21) (network reciprocity). Note that the last two

terms in each of equations (2.2)–(2.3), (2.7)–(2.8), (2.10)–(2.11), (2.13)–(2.14), (2.16)–(2.17) and (2.20)–(2.21) are equal because equations (1.4) and (1.5) are established by definition. Equations (2.2)–(2.3), (2.7)–(2.8), (2.10)–(2.11), (2.13)–(2.14), (2.16)–(2,17) and (2.20)–(2.21) suggest a conversion rule of representation, i.e. $(D'_r, D'_g) \rightarrow (D'_{r\,rev}, D'_{g\,rev})$. Note that these calculations of coordination transformation by five rules are detailed within our previous work (i.e. equations (2.6)–(2.21)) [15].

The shift in new coordinates $(D'_{r\,rev}, D'_{g\,rev})$ by the DUF is different from that under all the previous five rules. Unlike Nowak's five rules, the DUF simultaneously relaxes the GID and enhances the RAD (figure 1): the DUF does not unilaterally enhance the negative value of dilemma strength (figure 1*b* and *c*). Moreover, the introduction of the DUF does not change the game class. By contrast, the five reciprocity rules can cause three types of changes in game class by shifting the origin of the coordinates: (i) PD to chicken, (ii) PD to SH, and (iii) PD to trivial (no dilemmas). The DUF is unique in the enhancement of dilemma strength and is the only rule that can cause enhancement of the RAD while relaxing the GID without changes in the origin. No other rules lead to the enhancement of any dilemma upon introduction.

# 3. Analysis

We combine the concept of DUF with the five reciprocity rules; thus, the player, considering their current wealth, plays a game in which one of the five reciprocity rules works. Each combination of reciprocity mechanisms is calculated as follows.

## 3.1. Combining dynamic utility function and direct reciprocity

$$\begin{pmatrix} \log\left(\dfrac{\{R/(1-\lambda)\}+w}{w}\right) & \log\left(\dfrac{\{S+\lambda P/(1-\lambda)\}+w}{w}\right) \\ \log\left(\dfrac{\{T+\lambda P/(1-\lambda)\}+w}{w}\right) & \log\left(\dfrac{\{P/(1-\lambda)\}+w}{w}\right) \end{pmatrix}. \tag{3.1}$$

Again, $\lambda$ is the probability of two players meeting each other in another round, and $w$ is the current wealth of a player. The coordinates $(D'_r, D'_g)$ are transferred to $(D'_{r\,rev2}, D'_{g\,rev2})$ by DUF and direct reciprocity as follows (figure 2*b* and *c*):

$$D'_{g\,rev2} = \frac{\log\left(\left(\{T+\lambda P/(1-\lambda)\}+w\right)/w\right) - \log\left(\left(\{R/(1-\lambda)\}+w\right)/w\right)}{\log\left(\left(\{R/(1-\lambda)\}+w\right)/w\right) - \log\left(\left(\{P/(1-\lambda)\}+w\right)/w\right)} = f_{DUF\&DR}(T,R,P)$$
$$= f_{DUF\&DR}(D'_g, R, P) \tag{3.2}$$

and

$$D'_{r\,rev2} = \frac{\log\left(\left(\{P/(1-\lambda)\}+w\right)/w\right) - \log\left(\left(\{S+\lambda P/(1-\lambda)\}+w\right)/w\right)}{\log\left(\left(\{R/(1-\lambda)\}+w\right)/w\right) - \log\left(\left(\{P/(1-\lambda)\}+w\right)/w\right)} = g_{DUF\&DR}(S,R,P)$$
$$= g_{DUF\&DR}(D'_r, R, P). \tag{3.3}$$

## 3.2. Combining dynamic utility function and indirect reciprocity

$$\begin{pmatrix} \log\left(\dfrac{R+w}{w}\right) & \log\left(\dfrac{\{(1-q)S+qP\}+w}{w}\right) \\ \log\left(\dfrac{\{(1-q)T+qP\}+w}{w}\right) & \log\left(\dfrac{P+w}{w}\right) \end{pmatrix}. \tag{3.4}$$

Again, $q$ is the probability of knowing the reputation of another individual, and $w$ is the current wealth of a player. The coordinates $(D'_r, D'_g)$ are transferred to $(D'_{r\,rev2}, D'_{g\,rev2})$ by DUF and indirect reciprocity as follows (figure 2*b* and *c*):

$$D'_{g\,rev2} = \frac{\log\left(\dfrac{\{(1-q)T+qP\}+w}{w}\right) - \log\left(\dfrac{R+w}{w}\right)}{\log\left(\dfrac{R+w}{w}\right) - \log\left(\dfrac{P+w}{w}\right)} = f_{DUF\&IR}(T,R,P) = f_{DUF\&IR}(D'_g, R, P) \tag{3.5}$$

and

$$D'_{r\,\text{rev2}} = \frac{\log\left(\dfrac{P+w}{w}\right) - \log\left(\dfrac{\{(1-q)S+qP\}+w}{w}\right)}{\log\left(\dfrac{R+w}{w}\right) - \log\left(\dfrac{P+w}{w}\right)} = g_{DUF\&IR}(S,R,P) = g_{DUF\&IR}(D'_r,R,P). \quad (3.6)$$

Again, direct reciprocity and indirect reciprocity of the same strength will transfer the phase plane to the same coordinates [25].

## 3.3. Reciprocity mechanism combining dynamic utility function and kin selection

$$\begin{pmatrix} \log\left(\dfrac{R+w}{w}\right) & \log\left(\dfrac{\{(S+rT)/(1+r)\}+w}{w}\right) \\ \log\left(\dfrac{\{(T+rS)/(1+r)\}+w}{w}\right) & \log\left(\dfrac{P+w}{w}\right) \end{pmatrix}. \quad (3.7)$$

Again, $r$ is the average relatedness between interacting individuals, and $w$ is the current wealth of a player. The coordinates $(D'_r, D'_g)$ are transferred to $(D'_{r\,\text{rev2}}, D'_{g\,\text{rev2}})$ by DUF and kin selection as follows (figure 2e and f):

$$D'_{g\,\text{rev2}} = \frac{\log\left((\{(T+rS)/(1+r)\}+w)/w\right) - \log\left((R+w)/w\right)}{\log\left((R+w)/w\right) - \log\left((P+w)/w\right)} = f_{DUF\&KS}(T,R,P) = f_{DUF\&KS}(D'_g,R,P) \quad (3.8)$$

and

$$D'_{r\,\text{rev2}} = \frac{\log\left((P+w)/w\right) - \log\left((\{(S+rT)/(1+r)\}+w)/w\right)}{\log\left((R+w)/w\right) - \log\left((P+w)/w\right)} = g_{DUF\&KS}(S,R,P) = g_{DUF\&KS}(D'_r,R,P). \quad (3.9)$$

## 3.4. Combining dynamic utility function and group selection

$$\begin{pmatrix} \log\left(\dfrac{(m+n)R+w}{w}\right) & \log\left(\dfrac{(nS+mR)+w}{w}\right) \\ \log\left(\dfrac{(nT+mP)+w}{w}\right) & \log\left(\dfrac{(m+n)P+w}{w}\right) \end{pmatrix}. \quad (3.10)$$

Again, $m$ is the number of groups, $n$ is the maximum size of a group and $w$ is the current wealth of a player. The coordinates $(D'_r, D'_g)$ are transferred to $(D'_{r\,\text{rev2}}, D'_{g\,\text{rev2}})$ by DUF and group selection as follows (figure 3b and c):

$$D'_{g\,\text{rev2}} = \frac{\log\left(((nT+mP)+w)/w\right) - \log\left(((m+n)R+w)/w\right)}{\log\left(((m+n)R+w)/w\right) - \log\left(((m+n)P+w)/w\right)} = f_{DUF\&GS}(T,R,P) = f_{DUF\&GS}(D'_g,R,P) \quad (3.11)$$

and

$$D'_{r\,\text{rev2}} = \frac{\log\left(\dfrac{(m+n)P+w}{w}\right) - \log\left(\dfrac{(nS+mR)+w}{w}\right)}{\log\left(\dfrac{(m+n)R+w}{w}\right) - \log\left(\dfrac{(m+n)P+w}{w}\right)} = g_{DUF\&GS}(S,R,P) = g_{DUF\&GS}(D'_r,R,P). \quad (3.12)$$

## 3.5. Combining dynamic utility function and network reciprocity

$$\begin{pmatrix} \log\left(\dfrac{R+w}{w}\right) & \log\left(\dfrac{S+H+w}{w}\right) \\ \log\left(\dfrac{T-H+w}{w}\right) & \log\left(\dfrac{P+w}{w}\right) \end{pmatrix}. \quad (3.13)$$

Again, $k$ is the number of neighbours and $w$ is the current wealth of a player. The coordinates $(D'_r, D'_g)$ are transferred to $(D'_{r\,\text{rev2}}, D'_{g\,\text{rev2}})$ by DUF and network reciprocity as follows (figure 3e and f):

$$D'_{g\,\text{rev2}} = \frac{\log\left((T-H+w)/w\right) - \log\left((R+w)/w\right)}{\log\left((R+w)/w\right) - \log\left((P+w)/w\right)} = f_{DUF\&NR}(T,R,P) = f_{DUF\&NR}(D'_g,R,P) \quad (3.14)$$

and

$$D'_{r\,\text{rev2}} = \frac{\log\left((P+w)/w\right) - \log\left((S+H+w)/w\right)}{\log\left((R+w)/w\right) - \log\left((P+w)/w\right)} = g_{DUF\&NR}(S,R,P) = g_{DUF\&NR}(D'_r,R,P). \quad (3.15)$$

$D'_{r\,\text{rev2}}$ and $D'_{g\,\text{rev2}}$ are derived as equations (3.2)–(3.3) (DUF & direct reciprocity), equations (3.5)–(3.6) (DUF & indirect reciprocity), equations (3.8)–(3.9) (DUF & kin selection), equations (3.11)–(3.12) (DUF & group selection) and equations (3.14)–(3.15) (DUF & network reciprocity). Again, the last two terms in each of equations (3.2)–(3.3), (3.5)–(3.6), (3.8)–(3.9), (3.11)–(3.12) and (3.14)–(3.15) are equal because equations (1.4) and (1.5) are established by definition. Equations (3.2)–(3.3), (3.5)–(3.6), (3.8)–(3.9), (3.11)–(3.12) and (3.14)–(3.15) suggest a conversion rule of representation, i.e. $(D'_r,\ D'_g) \rightarrow (D'_{r\,\text{rev2}},\ D'_{g\,\text{rev2}})$.

# 4. Discussion

The current study is very similar to our previous study [15]. However, this is distinctively different in the findings. DUF is a dynamic version of the utility function, whereas the traditional utility function assumes the independence from current wealth, that is, a static model. Game theory by its definition should be dynamic as long as players repeat games. In this sense, the five rules should be fundamentally viewed not under the static utility functions, but under DUF. We here call the DUF the sixth reciprocity mechanism because it modifies the elements of a pay-off matrix, as the Nowak's five rules do. However, we should note that the current DUF model is not a functional mechanism, unlike the Nowak's five rules, but a more realistic model considering the effects of current wealth in the optimization of individual behaviour. We here show that DUF changes the traditional view of dilemma structure that has been assumed under static, or quasi-static model of the von Neumann–Morgenstern axioms. Thus, we showed that the dilemma structures under DUF are what we have to look for when we consider all other dilemma relaxation rules.

We analysed the dilemma strength of a game with a sixth reciprocity mechanism, a DUF, compared with that of a game with Nowak's five rules. The current result explains why the DUF promotes cooperative behaviour only in a chicken game [17]. RAD enhancement by the DUF means that the DUF strengthens the dilemma under the SH game (figure 1). We should also note that the coordinates change in the chicken and trivial regions in the DUF (see the grids of these regions in figure 1b and c). An increase in the wealth level $w$ in the DUF decreases the degree of relaxation/enhancement towards the default (figure 1). This result is consistent with previous analysis of a dynamic game (with a DUF) that becomes a static game as wealth $w \rightarrow \infty$ [17]. These properties of the DUF are distinct from those of Nowak's five rules of relaxation [15,16].

Under the DUF, the GID is relaxed and the RAD is amplified simultaneously in the same game structure (figure 1b and c). This fact is intuitive because the GID relegates humans to defective behaviours that deviate from cooperative actions more than the RAD does in social contexts. Note that the GID is inspired by an ambitious intention to exploit others more seriously, while the RAD is caused by the fear of being exploited by others. In other words, the GID is an indicator of the intention of exploitation, while the RAD is an indicator of the avoidance of exploitation [9]. Therefore, the relaxation of the GID is more critical to the development of cooperative behaviour. In this sense, the DUF should have played an important role in the evolution of animal and human societies.

We assume that the pay-off (utility) matrix depends on the player's current state, but more realistically, we can expect that the pay-off matrix also depends on the current state of opponent [24,26]. We also currently assume that $r_t$ are independent, identically distributed random variables (i.e. i.i.d.r.v.). In future, this condition may be relaxed, for example, depending on the current wealth, because growth rates are more likely to be depending on it. This random variable is more likely to be dependent on the current wealth. However, any changes in the current conditions remain to be unsolved, invoking the difficulty in analytic derivations.

The concept of the DUF, which considers the player's current wealth, can be combined with five reciprocity rules. All combinations of the DUF and five rules work effectively to relax the dilemma strength (figures 2, 3 and 4). In particular, the GID is dramatically relaxed by a combination of reciprocity mechanisms because both the DUF and the five rules relax the GID. By contrast, the DUF enhances the RAD; thus, the combinations of the DUF and five rules cannot be expected to relax the RAD. If the effect of the DUF is strong (i.e. the current wealth of player is small), the RAD may be enhanced by the DUF, despite RAD relaxation according to the five rules. However, as mentioned above, the GID is a more critical obstacle than the RAD for the promotion of cooperation. Therefore,

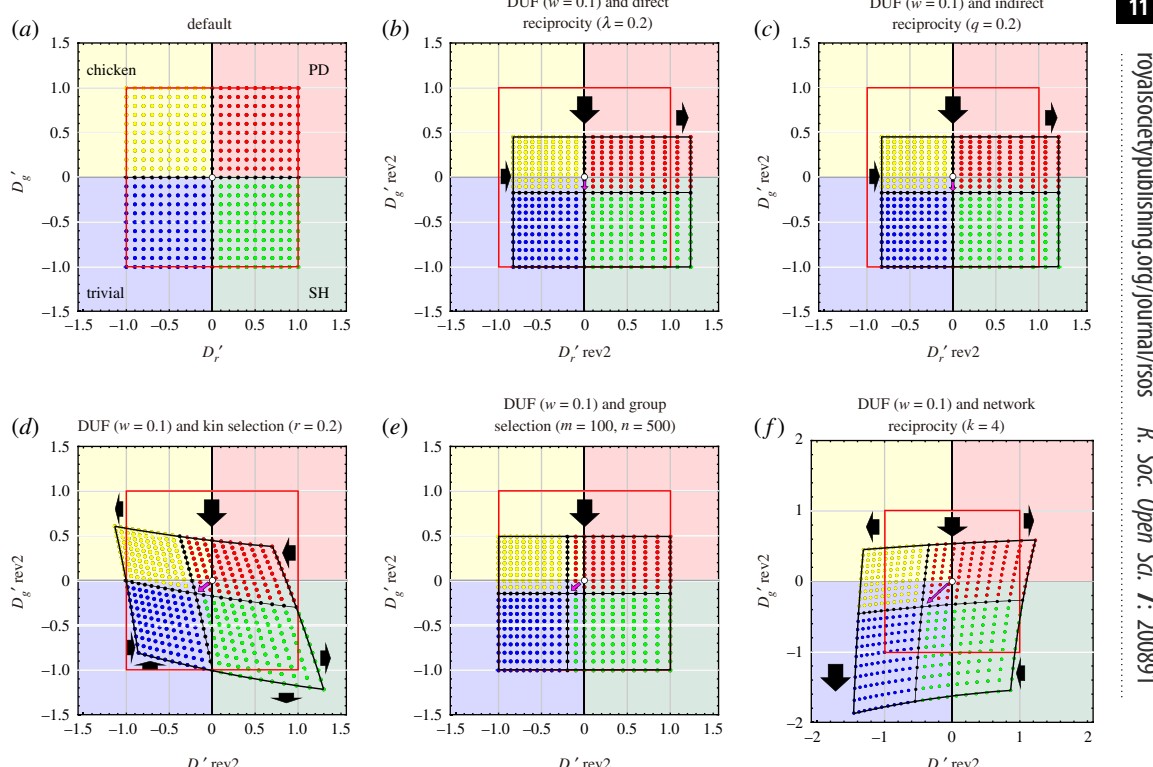

**Figure 4.** Transformation of the default phase plane ($-1 \leq D'_g, D'_r \leq +1$) with the introduction of the combined reciprocity mechanism of all five rules and the DUF in all $2 \times 2$ games. The shaded background colours indicate the regions of trivial (blue), prisoner's dilemma (PD) (red), chicken (yellow) and stag-hunt (SH) (green) games. The default game classes for each coordinate are indicated by the colour of the dots. (a) Default phase plane. (b–f) Transformed phase plane with the introduction of the DUF and (b) direct reciprocity, (c) indirect reciprocity, (d) kin selection, (e) group selection and (f) network reciprocity. The origin in the default (white circle) moves in the direction of the points indicated by pink arrows. The red squares indicate the default phase planes. The thick black arrows indicate the relaxation and enhancement of the GID and RAD. The basic parameters of the pay-off matrix are fixed as $R = 3$ and $P = 2$.

the combination of the DUF and the five reciprocity mechanisms is a highly effective promotion mechanism of cooperative behaviour in pairwise games.

The concept of the DUF is a possible alternative framework to the five reciprocity protocols elucidated by Nowak, leading to the evolution of mutual cooperation. More importantly, any games played by human and animal societies are dynamic [17,24]. Therefore, the current DUF should apply to any game in any society. This universality means that a game with the DUF is a true dynamic game that should follow the canonical form of games.

Ethics. The authors confirm that the study did not use humans or animals.

Data accessibility. The authors confirm that the article has no data.

Authors' contribution. H.I. and J.T. conceived the study. H.I. generated the figures. H.I. and J.T. wrote the manuscript.

Competing interests. The authors declare that they have no competing interests.

Funding. This work was supported by the JSPS KAKENHI (grant nos. 17J06741 and 17H04731 to H.I., and 18K18924 and 19KK0262 to J.T.).

Acknowledgements. We thank Prof. Jin Yoshimura for valuable feedback and discussions.

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
