## [Reviewer comments · Royal Society Open Science]

Review History

RSOS-200891.R0 (Original submission)

Review form: Reviewer 1

Is the manuscript scientifically sound in its present form?

Yes

Are the interpretations and conclusions justified by the results?

Yes

Is the language acceptable?

Yes

Do you have any ethical concerns with this paper?

No

Have you any concerns about statistical analyses in this paper?

No

Recommendation?

Accept with minor revision (please list in comments)

Comments to the Author(s)

In "Dynamic utility: the sixth reciprocity mechanism for the evolution of cooperation", Ito et al. propose and study the effect of novel reciprocity mechanism dynamic utility function in promotion of cooperation. They claim the universality of this rule as this mechanism can be applied to any game. The idea is interesting and I enjoyed reading the manuscript and fits perfectly to the domain of the Royal Society Open Science Journal. I warmly recommend publication, but some of my remarks are as follows and I want the authors to clarify a few points before acceptance.

1. In page 5, I don't understand that why is r_i taken as identically distributed for all decision makers? It should be more realistic if the random variables are not identical as per my understanding. Please comment.
2. The authors repeatedly use the previous study "Ito H, Tanimoto J. 2018 Scaling the phase-planes of social dilemma strengths shows game-class changes in the five rules governing the evolution of cooperation. R. Soc. Open Sci. 5, 181085. (doi:10.1098/rsos.181085)". The reader has to read this article in order to understand the actual link between this earlier study and the current manuscript. This is not suitable for academic papers. The article should advertise the main points of the findings and must describe to the non-specialist readers the significance of the research problem studied and the importance of the results.
3. The quality of figures should be improved. Individually, the figures are okay, but together they do not make a very coherent impression. Fig 1C should be Figure 1B as per description in the manuscript. All the figures can and should be made larger.
4. Please explain T, R, P, S in the pay-off matrix for non-specialist readers.
5. What does indicate the word 'poor players'?
6. In page 8, thrice it is written mathematically 'tends to max'. But, 'max' is a function, so it is not clear that what is the argument of max. What does this mean? max is a function so w_T tends to max of what argument? Is they try to say w_T is maximized?
7. In most of the equations, they use same notions 'f' and 'g' for all the functions (SEE equations 8,9,13,14,16,17,19,20,22,23,26,27,29,30,32,33,35,36,38,39,41,42). This creates confusion. Please use different notations to express distinct functions.
8. Some related works should be considered, for example: 1. doi.org/10.3390/e22040485, and 2. DOI: 10.1140/epjb/e2015-60270-7.

Review form: Reviewer 2

Is the manuscript scientifically sound in its present form?

Yes

Are the interpretations and conclusions justified by the results?

Yes

Is the language acceptable?

Yes

Do you have any ethical concerns with this paper?

No

Have you any concerns about statistical analyses in this paper?

No

Recommendation?

Accept with minor revision (please list in comments)

Comments to the Author(s)

I cast no doubt on the scientific soundness and novelty of this manuscript "Dynamic utility: the sixth reciprocity mechanism for the evolution of cooperation". My question is directed to the strong conclusion drawn from the formal analysis presented here, which is stressed in the title and the abstract; the sixth reciprocity mechanism.

Is "Dynamic utility" a MECHANISM which may be listed along with Nowak's five mechanisms including kin selection and others?

To my understanding, a theory in terms of dynamic (or any sort of) utility provides different ways of looking at the same phenomenon (behavior). As such, therefore, there is no new additional mechanism implied in it. For instance, we can imagine a group of animals in which kin selection is strongly relevant and another animal group with no or negligible kin selection. Comparing these two groups leads us to argue for the genuine effects due to the kin selection mechanism. However, comparison between cases with and without consideration of utility, as made in the manuscript, only brings us to the conclusion that the former is right and the latter is wrong. At least the proponent of utility would argue so while the opponent would disagree with such claim. Anyway this indicates that utility itself does not imply any objective mechanism but it gives just a good (or realistic) description. The authors should elaborate on why (in what sense) it is called "mechanism" before comparing it with the five mechanisms, each of which in fact requires special physical or biological circumstance for it to operate. Comparison between the cases with and without it, as made in the manuscript, only gives an impression of formal similarity, which is not strong enough to call it the sixth mechanism because if so any formal transformation can be a new mechanism! I believe that the last sentence of the Abstract undermines the strong claim of the authors: If "this sixth rule is universally ... applicable" then it means it is always there. It is not a new mechanism or otherwise a different kind from the five mechanisms.

Decision letter (RSOS-200891.R0)

Dear Dr Ito

On behalf of the Editors, I am pleased to inform you that your Manuscript RSOS-200891 entitled "Dynamic utility: the sixth reciprocity mechanism for the evolution of cooperation" has been accepted for publication in Royal Society Open Science subject to minor revision in accordance with the referee suggestions. Please find the referees' comments at the end of this email.

The reviewers and handling editors have recommended publication, but also suggest some minor revisions to your manuscript. Therefore, I invite you to respond to the comments and revise your manuscript.

- Ethics statement

- Data accessibility

<http://datadryad.org/submit?journalID=RSOS&manu=RSOS-200891>

- Competing interests

- Authors' contributions

- Acknowledgements

- Funding statement

Because the schedule for publication is very tight, it is a condition of publication that you submit the revised version of your manuscript before 09-Jul-2020. Please note that the revision deadline will expire at 00.00am on this date. If you do not think you will be able to meet this date please let me know immediately.

If your manuscript is newly submitted and subsequently accepted for publication, you will be asked to pay the article processing charge, unless you request a waiver and this is approved by Royal Society Publishing. You can find out more about the charges at

<https://royalsocietypublishing.org/rsos/charges>. Should you have any queries, please contact openscience@royalsociety.org.

on behalf of Professor Matjaz Perc (Associate Editor) and Mark Chaplain (Subject Editor)
openscience@royalsociety.org

Reviewer comments to Author:
Reviewer: 1

Comments to the Author(s)

In "Dynamic utility: the sixth reciprocity mechanism for the evolution of cooperation", Ito et al. propose and study the effect of novel reciprocity mechanism dynamic utility function in promotion of cooperation. They claim the universality of this rule as this mechanism can be applied to any game. The idea is interesting and I enjoyed reading the manuscript and fits perfectly to the domain of the Royal Society Open Science Journal. I warmly recommend publication, but some of my remarks are as follows and I want the authors to clarify a few points before acceptance.

1. In page 5, I don't understand that why is $\$r_i\$$ taken as identically distributed for all decision makers? It should be more realistic if the random variables are not identical as per my understanding. Please comment.
2. The authors repeatedly use the previous study "Ito H, Tanimoto J. 2018 Scaling the phase-planes of social dilemma strengths shows game-class changes in the five rules governing the evolution of cooperation. R. Soc. Open Sci. 5, 181085. (doi:10.1098/rsos.181085)". The reader has to read this article in order to understand the actual link between this earlier study and the current manuscript. This is not suitable for academic papers. The article should advertise the main points of the findings and must describe to the non-specialist readers the significance of the research problem studied and the importance of the results.
3. The quality of figures should be improved. Individually, the figures are okay, but together they do not make a very coherent impression. Fig 1C should be Figure 1B as per description in the manuscript. All the figures can and should be made larger.
4. Please explain T, R, P, S in the pay-off matrix for non-specialist readers.
5. What does indicate the word 'poor players'?
6. In page 8, thrice it is written mathematically 'tends to max'. But, 'max' is a function, so it is not clear that what is the argument of max. What does this mean? max is a function so w_T tends to max of what argument? Is they try to say w_T is maximized?

7. In most of the equations, they use same notions 'f' and 'g' for all the functions (SEE equations 8,9,13,14,16,17,19,20,22,23,26,27,29,30,32,33,35,36,38,39,41,42). This creates confusion. Please use different notations to express distinct functions.

8. Some related works should be considered, for example: 1. doi.org/10.3390/e22040485, and 2. DOI: 10.1140/epjb/e2015-60270-7.

Reviewer: 2

Comments to the Author(s)

I cast no doubt on the scientific soundness and novelty of this manuscript "Dynamic utility: the sixth reciprocity mechanism for the evolution of cooperation". My question is directed to the strong conclusion drawn from the formal analysis presented here, which is stressed in the title and the abstract; the sixth reciprocity mechanism.

Is "Dynamic utility" a MECHANISM which may be listed along with Nowak's five mechanisms including kin selection and others?

To my understanding, a theory in terms of dynamic (or any sort of) utility provides different ways of looking at the same phenomenon (behavior). As such, therefore, there is no new additional mechanism implied in it. For instance, we can imagine a group of animals in which kin selection is strongly relevant and another animal group with no or negligible kin selection. Comparing these two groups leads us to argue for the genuine effects due to the kin selection mechanism. However, comparison between cases with and without consideration of utility, as made in the manuscript, only brings us to the conclusion that the former is right and the latter is wrong. At least the proponent of utility would argue so while the opponent would disagree with such claim. Anyway this indicates that utility itself does not imply any objective mechanism but it gives just a good (or realistic) description. The authors should elaborate on why (in what sense) it is called "mechanism" before comparing it with the five mechanisms, each of which in fact requires special physical or biological circumstance for it to operate. Comparison between the cases with and without it, as made in the manuscript, only gives an impression of formal similarity, which is not strong enough to call it the sixth mechanism because if so any formal transformation can be a new mechanism! I believe that the last sentence of the Abstract undermines the strong claim of the authors: If "this sixth rule is universally ... applicable" then it means it is always there. It is not a new mechanism or otherwise a different kind from the five mechanisms.

Author's Response to Decision Letter for (RSOS-200891.R0)

See Appendix A.

Decision letter (RSOS-200891.R1)

Dear Dr Ito,

It is a pleasure to accept your manuscript entitled "Dynamic utility: the sixth reciprocity mechanism for the evolution of cooperation" in its current form for publication in Royal Society Open Science.

on behalf of Professor Matjaz Perc (Associate Editor) and Mark Chaplain (Subject Editor)
openscience@royalsociety.org

Appendix A

Dear Prof. Matjaz Perc (Associate Editor) and Prof. Mark Chaplain (Subject Editor)

I would like to thank you for your time effort in reviewing our manuscript, and accepting our paper on Royal Society Open Science.

In this response letter, *point-by-point responses to the comments and suggestions are shown in bold italic font*. Red text indicates the addition of text that has been added to the revised manuscript. I hope the revisions made have ensured the manuscript is now satisfactory for publication in Royal Society Open Science.

Hiromu ITO

Reviewer comments to Author:

Reviewer: 1

Comments to the Author(s)

In "Dynamic utility: the sixth reciprocity mechanism for the evolution of cooperation", Ito et al. propose and study the effect of novel reciprocity mechanism dynamic utility function in promotion of cooperation. They claim the universality of this rule as this mechanism can be applied to any game. The idea is interesting and I enjoyed reading the manuscript and fits perfectly to the domain of the Royal Society Open Science Journal. I warmly recommend publication, but some of my remarks are as follows and I want the authors to clarify a few points before acceptance.

REPLY:

Thank you for finding our topic interesting, and we sincerely thank you for your recommendation.

1. In page 5, I don't understand that why is r_i taken as identically distributed for all decision makers? It should be more realistic if the random variables are not identical as per my understanding. Please comment.

REPLY:

Thanks for this comment. We are very surprised with this comment. In reality, the conditions for r_i may very well depend on the current wealth. For example, the growth rate of wealth should be different with us and Bill Gates or Queen Elizabeth. In the current settings, we suppose that the probability distribution of r_i should be independent on previous states as in coin flipping or a throwing dice. We are looking for its dependence on the current wealth.

But we could not find this solution from our stochastic analyses. So we follow the traditional development started by Jin Yoshimura (Yoshimura and Clark 1991 for geometric mean fitness and Yoshimura et al. 2013a,b J. Ethol.. We add a short note on this point, so that the reader may tackle this problem. We want to see this solution in future.

We add the explanation that you pointed out.

First paragraph in page 18.

We assume that the payoff (utility) matrix depends on the player's current state, but more realistically, we can expect that the payoff matrix also depends on the current state of opponent [24,26]. We also currently assume that r_t are independent, identically-distributed random variables, (i.e., i.i.d.r.vs.). In future, this condition may be relaxed, for example, depending on the current wealth, because growth rates are more likely to be depending on it. This random variable is more likely to be dependent on the current wealth. However, any changes in the current conditions remains to be unsolved, invoking the difficulty in analytic derivations.

2. The authors repeatedly use the previous study "Ito H, Tanimoto J. 2018 Scaling the phase-planes of social dilemma strengths shows game-class changes in the five rules governing the evolution of cooperation. R. Soc. Open Sci. 5, 181085. (doi:10.1098/rsos.181085)". The reader has to read this article in order to understand the actual link between this earlier study and the current manuscript. This is not suitable for academic papers. The article should advertise the main points of the findings and must describe to the non-specialist readers the significance of the research problem studied and the importance of the results.

REPLY:

Thanks for this comment. When we submitted this paper, we were repeatedly pointed out by the Editorial Office that it was difficult to understand the difference from previous study (doi: 10.1098/rsos.181085). Therefore, this paper emphasizes the differences from previous study. We expect that the concept of the dilemma phase-plane will be well understood in the explanation from page 4 to the middle of 5 in background section. In addition, we show the formula for calculating the dilemma strength for all five reciprocity rules (Eq. 12 to 27), thus, we think the current manuscript contains enough information about our previous work. We added explanation about main points and findings of the current study as follows.

First paragraph in discussion section.

The current study is very similar with our previous study [15]. However, this is distinctively different in the findings. DUF is a dynamic version of the utility function, whereas the traditional utility function assumes the independence from current wealth, that is, a static model. Game theory by its definition should be dynamic as long as players repeat games. In

this sense, the five rules should be fundamentally viewed not under the static utility functions, but under DUF. We here call the DUF the sixth reciprocity mechanism because it modifies the elements of a payoff matrix, as the Nowak's five rules do. However, we should note that the current DUF model is not a functional mechanism unlike the Nowak's five rules, but a more realistic model considering the effects of current wealth in the optimization of individual behavior. We here show that DUF changes the traditional view of dilemma structure that has been assumed under static, or quasi-static model of the von Neumann-Morgenstern axioms. Thus, we showed that the dilemma structures under DUF is what we have to look for when we consider all other dilemma relaxation rules.

3. The quality of figures should be improved. Individually, the figures are okay, but together they do not make a very coherent impression. Fig 1C should be Figure 1B as per description in the manuscript. All the figures can and should be made larger.

REPLY:

Thank you for the advice. According to your advice, we delete Figure 1B for simplify.

The figure attached to the manuscript word file may be of poor image quality because it is a bitmap style, however, at the same time, we uploaded the EPS version figures. We think that the EPS data has sufficient image quality.

New figure 1.

4. Please explain T, R, P, S in the pay-off matrix for non-specialist readers.

REPLY:

Thanks for the suggestion. I add the explanation about each elements of payoff matrix.

Last paragraph in page 3.

This payoff matrix means as follows: if both cooperate, they receive the 'reward' R ; if both defect, they get 'punishment' P ; and if one chooses cooperation while the other defects, the defector gets the 'temptation' T and the cooperator left the payoff of 'sucker' S [11].

5. What does indicate the word 'poor players'?

REPLY:

Thanks for this comment and I apologize for my lack of explanation. In this manuscript, from the DUF perspective, 'poor player' is a player with a small current wealth w .

First paragraph in page 7.

However, we cannot explain why cooperation is promoted by poor players (whose current wealth, w , are very low) in the chicken game but not in the PD game or SH game [17].

6. In page 8, thrice it is written mathematically 'tends to max'. But, 'max' is a function, so it is not clear that what is the argument of max. What does this mean? max is a function so w_T tends to max of what argument? Is they try to say w_T is maximized?

REPLY:

Thanks for this comment. We apologize for our lack of explanation. We have added a more detailed explanation as follows.

Last paragraph in page 8.

By introducing dynamic utility, i.e., the sixth reciprocity mechanism, the coordinates (D_r' , D_g') of the default game are transferred to the new coordinates ($D_r'_{rev}$, $D_g'_{rev}$) (Fig. 1). The DUF-transformed diagram depends on the current wealth level w (Fig. 1B and 1C: w -dependence of the new diagram). Mathematically, if current wealth w is small, the distortion of the dilemma structure is very large compared with the static model. But when $w \rightarrow \infty$ (means a person becomes extraordinary rich), this distortion disappears and converges to the solution for the static model:

$$D_g'_{rev} = \lim_{w \rightarrow \infty} \left[\frac{\log[(T+w)/(R+w)]}{\log[(R+w)/(P+w)]} \right] = \lim_{w \rightarrow \infty} \left[\frac{\log\left(1+\frac{T}{w}\right) - \log\left(1+\frac{R}{w}\right)}{\log\left(1+\frac{R}{w}\right) - \log\left(1+\frac{P}{w}\right)} \right] \cong \frac{\frac{T}{w} - \frac{R}{w}}{\frac{R}{w} - \frac{P}{w}} = \frac{T-R}{R-P} = D_g' \quad (10)$$

$$D_r'_{rev} = \lim_{w \rightarrow \infty} \left[\frac{\log[(P+w)/(S+w)]}{\log[(R+w)/(P+w)]} \right] = \lim_{w \rightarrow \infty} \left[\frac{\log\left(1+\frac{P}{w}\right) - \log\left(1+\frac{S}{w}\right)}{\log\left(1+\frac{R}{w}\right) - \log\left(1+\frac{P}{w}\right)} \right] \cong \frac{\frac{P}{w} - \frac{S}{w}}{\frac{R}{w} - \frac{P}{w}} = \frac{P-S}{R-P} = D_r' \quad (11)$$

Thus, the static model is considered an approximate model of the DUF games for extremely rich people, but not for the ordinary people.

7. In most of the equations, they use same notions 'f' and 'g' for all the functions (SEE equations 8,9,13,14,16,17,19,20,22,23,26,27,29,30,32,33,35,36,38,39,41,42). This creates confusion. Please use different notations to express distinct functions.

REPLY:

Thanks for this comment. We add subscript on f and g for all equations mentioned above. For example, the equation (8) and (9) is modified as follows:

$$D_g'_{rev} = \frac{\log\left(\frac{T+w}{w}\right) - \log\left(\frac{R+w}{w}\right)}{\log\left(\frac{R+w}{w}\right) - \log\left(\frac{P+w}{w}\right)} = \frac{\log[(T+w)/(R+w)]}{\log[(R+w)/(P+w)]} = f_{DUF}(T, R, P) = f_{DUF}(D_g', R, P) \quad (8)$$

$$D_r'_{rev} = \frac{\log\left(\frac{P+w}{w}\right) - \log\left(\frac{S+w}{w}\right)}{\log\left(\frac{R+w}{w}\right) - \log\left(\frac{P+w}{w}\right)} = \frac{\log[(P+w)/(S+w)]}{\log[(R+w)/(P+w)]} = g_{DUF}(S, R, P) = g_{DUF}(D_r', R, P). \quad (9)$$

8. Some related works should be considered, for example: 1. doi.org/10.3390/e22040485, and 2. DOI: 10.1140/epjb/e2015-60270-7.

REPLY:

Thanks for the suggestion. Due to my lack of understanding, but I could only find a relation to our manuscript in DOI: 10.1140/epjb/e2015-60270-7. We add some sentences into discussion section and reference as follows.

First paragraph in page 18.

We assume that the payoff (utility) matrix depends on the player's current state, but more realistically, we can expect that the payoff matrix also depends on the current state of opponent [24,26]. We also currently assume that r_t are independent, identically-distributed random variables, (i.e., i.i.d.r.vs.). In future, this condition may be relaxed, for example, depending on the current wealth, because growth rates are more likely to be depending on it. This random variable is more likely to be dependent on the current wealth. However, any changes in the current conditions remains to be unsolved, invoking the difficulty in analytic derivations.

References

26. Wang Z, Wang L, Szolnoki A, Perc M. 2015 Evolutionary games on multilayer networks: a colloquim. Eur. Phys. J. B. 88, 124.

Reviewer: 2

Comments to the Author(s)

I cast no doubt on the scientific soundness and novelty of this manuscript "Dynamic utility: the sixth reciprocity mechanism for the evolution of cooperation". My question is directed to the strong conclusion drawn from the formal analysis presented here, which is stressed in the

title and the abstract; the sixth reciprocity mechanism.

Is "Dynamic utility" a MECHANISM which may be listed along with Nowak's five mechanisms including kin selection and others?

To my understanding, a theory in terms of dynamic (or any sort of) utility provides different ways of looking at the same phenomenon (behavior). As such, therefore, there is no new additional mechanism implied in it. For instance, we can imagine a group of animals in which kin selection is strongly relevant and another animal group with no or negligible kin selection. Comparing these two groups leads us to argue for the genuine effects due to the kin selection mechanism. However, comparison between cases with and without consideration of utility, as made in the manuscript, only brings us to the conclusion that the former is right and the latter is wrong. At least the proponent of utility would argue so while the opponent would disagree with such claim. Anyway this indicates that utility itself does not imply any objective mechanism but it gives just a good (or realistic) description. The authors should elaborate on why (in what sense) it is called "mechanism" before comparing it with the five mechanisms, each of which in fact requires special physical or biological circumstance for it to operate. Comparison between the cases with and without it, as made in the manuscript, only gives an impression of formal similarity, which is not strong enough to call it the sixth mechanism because if so any formal transformation can be a new mechanism! I believe that the last sentence of the Abstract undermines the strong claim of the authors: If "this sixth rule is universally ... applicable" then it means it is always there. It is not a new mechanism or otherwise a different kind from the five mechanisms.

REPLY:

You are very correct! We are very happy to see your comments/views. The traditional game theory based on the expected utility is a comparison model or at most a quasi-optimization model that maximizes the expected preference, but not an optimization of future wealth. All decision models including economics, management science and financial engineering (e.g., portfolio selection, BSM model, and investment strategy in general) are based on the future expectation as that of in the game theory. Thus, we need to shift from the current static/quasi-static model to a dynamic model. However, our scientific society takes sometimes a lot of time to digest a new theory/development. We add a few lines to discuss the usage of mechanisms as suggested by you. We hope our current vision becomes more popular. Thus, we are very, very happy to receive your comments.

We wrote new first paragraph in discussion section as below:

The current study is very similar with our previous study [15]. However, this is distinctively different in the findings. DUF is a dynamic version of the utility function, whereas the traditional utility function assumes the independence from current wealth, that is, a static model. Game theory by its definition should be dynamic as long as players repeat games. In this sense, the five rules should be fundamentally viewed not under the static utility functions, but under DUF. We here call the DUF the sixth reciprocity mechanism because it modifies the elements of a payoff matrix, as the Nowak's five rules do. However, we should note that the current DUF model is not a functional mechanism unlike the Nowak's five rules, but a more realistic model considering the effects of current wealth in the optimization of individual behavior. We here show that DUF changes the traditional view of dilemma structure that has been assumed under static, or quasi-static model of the von Neumann-Morgenstern axioms. Thus, we showed that the dilemma structures under DUF is what we have to look for when we consider all other dilemma relaxation rules.